# An Analysis of Gains to US Acquiring REIT Shareholders in Domestic and Cross-Border Mergers before and after the Subprime Mortgage Crisis

**Alan T. Wang [1], Yu-Hong Liu [1],\* and Yu-Chen Chang [2]**

[1] Institute of Finance, National Cheng Kung University, Tainan City 70101, Taiwan; wangt@mail.ncku.edu.tw
[2] JP Morgan, 8F No. 108, Sec. 5, Xinyi Road., Xinyi Dist., Taipei City 11047, Taiwan; a0981103127@gmail.com
\* Correspondence: yuhong@mail.ncku.edu.tw; Tel.: +886-6-2757575-53429; Fax: +886-6-2744104

**Abstract:** This paper examines the abnormal returns of acquiring real estate investment trusts (REITs) around the announcement of acquisitions before and after the subprime mortgage crisis. Based on 182 domestic and cross-border US REIT acquisition announcements from 2005 to 2010, the acquiring trusts experienced a 0.73% abnormal return, on average. When the sample was divided into pre-crisis, crisis, and after-crisis subsamples, the acquiring trusts enjoyed the largest abnormal returns (1.86%) for domestic acquisitions during the crisis period. Before the crisis, when the acquisition was cross-border, the target was private, or the transaction was cash-financed, the acquiring trust experienced larger abnormal returns. During the crisis period, the acquiring trust gained larger abnormal returns when the transaction value was larger. After the crisis period, the acquiring trust achieved less abnormal returns in cross-border mergers. For both pre- and after-crisis periods, the shareholders of the acquirer enjoyed larger abnormal returns when the mergers were cash-financed, regardless of whether the target was public or privately held. Neither the blockholder monitoring nor the signaling hypothesis can explain such value gains. The structural changes in the acquirer's abnormal returns are possibly due to the increased risk aversion of the market participants following the crisis.

**Keywords:** merger; subprime mortgage crisis; event study

## 1. Introduction

In the early period of the 2000s, the U.S. was experiencing mild economic growth and rising housing prices when the interest rates were low. Because of the rising home prices later on, bankers were encouraged to make more loans for higher compensations, and the quality of the mortgage loans started to deteriorate, especially when the mortgage rates started to rise. The first disruption of the credit market can be dated back to 7 August 2007, when the French bank BNP Paribas suspended the redemption of the shares held in some of its money market funds [1]. Furthermore, on 15 September 2018, the investment bank Lehman Brothers filed for the largest bankruptcy in U.S. history due to its loss during the subprime mortgage market. Reinhart and Rogoff [2] stated that the aftermath of the crisis has three characteristics: the asset market collapses are prolonged, the banking crisis is associated with profound declines in output and employment, and the level of government debt tends to explode. The outbreak of the global financial crisis in 2008 has reshaped the landscape of the financial markets as well. Investors have become more risk-averse, and that has changed the corporate strategies in business activities. It is important to examine how corporations have adjusted their mergers and acquisitions strategies since the crisis and to explore how market participants perceive the acquisition activities. To keep corporations sustainable, acquirers must

identify the right targets from long-term perspectives. For example, are the prices of the targets appropriate during the normal periods or the crisis periods? Will the acquisitions help the corporations grow? If the acquisitions are favorable, the market participants will revalue the acquirer's stock price.

Acquisitions may benefit shareholders from different sources, such as revenue enhancement from marketing gains and market monopoly power, cost reductions from economies of scale and the elimination of inefficient management, tax gains, and a reduced cost of capital. The wealth effect of the acquisition has been well documented in the corporate finance literature. A stylized fact concerning mergers is that the shareholders of bidding firms suffer wealth loss at the announcement of stock-financed merger transactions. This is consistent with the information asymmetry hypothesis, in which the manager considers the firm's stocks overvalued, and stock instead of cash is chosen as the payment method for a merger transaction. Jensen and Ruback [3] reported the short-run evidence from studies that used event studies and looked at the effect of a merger announcement on abnormal stock returns. The abnormal returns associated with successful corporate takeover bids for the target firms are 30%, 20%, and 8% in the cases of a tender offer, merger, and proxy contest, respectively. However, for the bidding firms, the abnormal returns are much smaller or close to zero.

On the other hand, other corporate finance literature has documented the opposite results for the acquirer's abnormal returns. For example, Asquith et al. [4] concluded that bidding firms gain during the 21 days before the announcements of merger bids. Bidders' abnormal returns are positively related to the relative size of the merger partners, and the gains around the announcement period are larger for successful mergers. They concluded that their findings are consistent with the value-maximization behavior of the management of the bidding firms. The inconclusive results documented in earlier studies for the gains or losses of bidding firms around the merger announcement are partially explained by the relative size of the merger partner and the time period of the merger.

The purpose of this study is to reexamine the effects of real estate investment trust (REIT) acquisitions on the wealth of the shareholders of the acquiring trust before and after the subprime mortgage crisis. REITs allow investors to indirectly invest in professionally managed commercial real estate portfolios and then distribute rents and capital gains to their investors. REITs have unique institutional settings characterized by very codified and transparent corporate governance. Since REITs do not normally pay federal income taxes and are required to distribute at least 90% of their taxable income, they are highly dependent on their ability to access external capital. Thus, REITs are especially vulnerable during a credit crisis [5]. There are several advantages for a REIT to acquire another trust. For example, net operating losses can be used to offset capital gains tax liabilities from the sale of trust property, making an existing trust an attractive target. Furthermore, the merger may replace existing inefficient management in the acquired trust and result in better utilization of assets (see Allen and Sirmans [6] for detailed descriptions of the institutional background of REITs).

The gains to the bidders from mergers when both the buyer and seller are REITs have also been examined in previous studies. The short-run evidence from the studies using event studies and looking at the effects of mergers on abnormal stock returns for REITs is also inconclusive. For example, with a sample of REIT mergers over the period of 1977–1983, Allen and Sirmans [6] concluded that REIT acquisitions significantly increased the wealth of the acquirer's shareholders, and they argued that the value gain comes from the improved management of the acquired trusts' assets, rather than the tax benefits.

Campbell et al. [7] examined the information content of the method of payment in REIT mergers from 1994 to 1998. They documented that, when the target firm is publicly held, the transactions are always stock-financed, and the acquiring firm's shareholders sustain small negative returns around the announcement date. The explanation for the negative returns is that the acquirer's stock is overvalued. When the target is privately held, the acquirer returns are positive in stock-financed mergers. Their finding may be explained by two hypotheses: the blockholder monitoring hypothesis and the signaling hypothesis. Chang [8] argued that this value enhancement may be caused by the monitoring benefits provided by new blockholders often observed in public-private mergers

(blockholder monitoring hypothesis). Alternatively, the owners of private targets are also expected to be better informed about the prospects of the acquiring firm, and their willingness to hold the acquirer's stock provides a positive signal to the market (signaling hypothesis). Campbell et al. [7] concluded that the information signaling hypothesis is the dominant explanation. Sahin [9] examined the performance of acquisitions in the REIT industry from 1994 to 1998. The results indicated that the acquiring REITs suffer statistically significant negative abnormal returns, while the target REITs earn statistically significant positive returns around the announcement date.

To further explore the wealth effect of REIT mergers on the acquiring trust, this study examines the short-run performance of the acquirer in 182 REIT mergers around the announcement dates in the period of 2005Q4–2010Q4 from the perspectives of domestic versus cross-border mergers. Unlike the previous studies, our sample period spans the subprime mortgage crisis period, and we try to highlight the importance of the change in the risk appetite of market participants due to the subprime mortgage crisis by distinguishing domestic from cross-border mergers.

Unlike previous REIT merger literature, there are considerably more REIT merger events in our sample. By considering domestic and cross-border REIT merger announcements over different subperiods, we found that the shareholders of the acquirer achieved significant value gains from cross-border mergers during the crisis period only. The gain is attributable to the lower stock prices of the target trusts. From a cross-sectional analysis, we found that, before the crisis period, if the merger was cross-border, the target trust was privately held, or the merger was cash-financed, the acquirer achieved larger abnormal returns. During the crisis period, a larger acquisition value was associated with a larger value gain to the acquirer. This finding reinforces the hypothesis that the gains come from the undervalued assets of the target REITs during the crisis period. Following the onset of the subprime mortgage crisis, the acquirer achieved more value gain when the target was domestic (rather than cross-border), the acquisition was cash-financed, or there were more states in which the acquirer had properties. This evidence suggests that investors increased their degree of risk aversion after the subprime mortgage crisis because cross-border acquisitions are associated with higher risk resulting from corporate governance, cultural differences, information asymmetries, and valuation issues. If the acquirer has properties in more states, the acquirer's sources of future cash flows are more geographically diversified. The remainder of this paper is organized as follows. Section 2 reviews the literature concerning the wealth effect in merger transactions. The data and research methodology are detailed in Section 3. An empirical analysis is given in Section 4. The final section concludes this paper.

## 2. Literature Review

### 2.1. Cross-Border Mergers

Internationalization theory suggests that cross-border acquisitions result in gains from diversification when a business seeks synergies from intangible assets, such as information-based assets [10,11]. Quah and Young [12] asserted that the management of both cultural and organizational integrations in cross-border mergers will tend to make acquisitions successful, but poor attention to these issues will destroy synergistic gains. Cross-border mergers are affected by factors such as cultural identity, physical distance, corporate governance, and equity market valuation differences [13]. In terms of valuation differences, if the difference is temporary, the cross-border acquisitions effectively arbitrage these differences. The valuation difference can also be permanent. Kindleberger [14] argued that cross-border acquisitions can occur because the expected earnings are larger or the cost of capital is lower. For example, if a firm is involved in overseas sales or imports, the firm can acquire a foreign target when the target currency depreciates.

With a sample of 4430 corporate acquisitions for the period of 1985–1995, Moeller and Schlingemann [15] found that US firms who acquire cross-border targets relative to those that acquire domestic targets suffer significantly lower announcement stock returns of approximately 1%. Moeller and Schlingemann [15] summarized several disadvantages of market integration, including increases

in competition in the market for corporate control, increases in hubris and agency problems, the cost of internalization, and a decrease in value from diversification.

Other evidence of bidders' abnormal stock returns in cross-border merger announcements is, however, mixed. Dutta et al. [16] found that bidders in the United States had positive cumulative abnormal returns in cross-border mergers. Martynova and Renneboog [17] documented that acquirers engaging in cross-border bids experienced fewer announcement effects than those associated with domestic acquisitions (0.4% and 0.6%, respectively). However, Aybar and Ficici [18] and Chakrabarti et al. [19] found negative cumulative abnormal returns.

## 2.2. REIT Mergers

Allen and Sirmans [6] conducted the first study concerning the wealth effects of REIT mergers by examining 38 successful REIT–REIT mergers from 1977 to 1983. They found significant positive abnormal returns for the acquirers, which is in contrast to the small negative return from corporate deals. Extending the research of Allen and Sirmans [6], McIntosh et al. [20] examined the return for 27 target REITs over the period of 1962–1986, finding a positive and significant average abnormal return of 2.16%. They concluded that the results are consistent with the hypothesis that target REITs achieve a positive wealth effect due to a merger announcement.

With a REIT merger sample over the period of 1994–1998, Campbell et al. [7] found that, when the target REIT was public, the transactions were always stock-financed, and the shareholders of the acquiring REITs suffered negative returns around the announcement. When the target REIT was privately held, cash financing, mixed financing, and the placement of the acquirer's stock with target owners were more prevalent. Acquirer shareholders achieved positive abnormal returns around the announcement of stock-financed mergers when the target was private, which is consistent with the monitoring by blockholders hypothesis and information signaling hypothesis. Sahin [9] also examined the performance of acquisitions in the REIT industry around the acquisition announcement. The results indicated that the acquiring REITs suffered statistically significant negative abnormal returns. This finding is in line with the research by Campbell et al. [7] but inconsistent with the finding of Allen and Sirmans [6]. The difference is argued to be due to the different environments in the 1980s and 1990s.

Ooi, Ong, and Neo [21] investigated 228 merger announcements in the Japanese and Singaporean REIT markets from 2002 to 2007, suggesting that aggressive growth acquisitions by Asian REITs were a result of improved economies of scale and better management practices. Their results showed that the bidding REITs earned positive and significant abnormal returns of 0.21%. Finally, Ling, and Petrova [22] examined the wealth effects of public-public and public-private REIT merger announcements from 1994 to 2007 and found that targets in public-private mergers earned higher abnormal returns than those in public-public announcements (cumulative abnormal returns were 10.38% and 7.7%, respectively).

## 2.3. Mergers in the Subprime Mortgage Crisis

Numerous studies examined the wealth effects of mergers during the subprime mortgage crisis period. Berger and Bouwman [23] indicated that healthy banks, particularly from the point of view of capital and liquidity, have an opportunity to improve their market share and profitability during a crisis by making acquisitions. Thus, this implies positive abnormal returns because acquirers can acquire other banks at lower prices and also benefit from portfolio diversification [24] and market power [25]. Furthermore, Reddy et al. [26] examined 26 countries' cross-border mergers from 2004 to 2010 and found that, following the onset of the crisis, companies in emerging countries took advantage of attractive asset prices by acquiring firms in developed countries. Beltratti and Paladino [27] also showed that investors attached significant uncertainty to the completion of deals and rewarded successful acquisitions with delayed abnormal returns during the crisis period.

## 3. Data and Methodology

### 3.1. Data

The sample consists of 182 merger announcements made by 229 publicly traded REITs on NYSE between 2005Q4 and 2010Q4. The data were collected from SDC Platinum and Datastream. Of the 182 merger announcements, there are 106 cash-financed, 4 stock-financed, and 10 hybrid transactions. For the rest of the transactions, they are indicated as "unknown" in SDC Platinum.

The 182 transaction announcements have a total value equivalent to 22062.59 million dollars. Table 1 illustrates the distribution of the number of REIT merger announcements for each year. Table 2 shows the compositions of REIT merger announcements for domestic and cross-border transactions before the subprime mortgage crisis period (2005Q4–2007Q1), during the subprime mortgage crisis period (2007Q2–2009Q1), and after the subprime mortgage crisis period (2009Q2–2010Q4).

**Table 1.** Number of acquisitions from 2005 to 2010 and corresponding transaction value.

| Year | Total Number of Acquisitions | Total Transaction Value (US$ million) |
|---|---|---|
| 2005 | 17 | $1817.22 |
| 2006 | 39 | $5165.38 |
| 2007 | 35 | $8817.75 |
| 2008 | 30 | $2201.90 |
| 2009 | 16 | $445.74 |
| 2010 | 45 | $3614.60 |
| Total | 182 | $2,2062.59 |

**Table 2.** Profile of domestic and cross-border mergers and acquisitions (M&A) announcements made before, during and after the subprime mortgage crisis.

| | Before the Crisis (1 October 2005– 31 March 2007) | During the Crisis (1 April 2007– 31 March 2009) | After the Crisis (1 April 2009– 31 December 2010) |
|---|---|---|---|
| Number (%) of domestic M&A | 56 (35%) | 54 (34%) | 50 (31%) |
| Number (%) of cross-border M&A | 9 (41%) | 7 (32%) | 6 (27%) |

### 3.2. Methodology

#### 3.2.1. Measuring Abnormal Returns

This study followed the standard event study methodology of Brown and Warner [28,29] to analyze the effect of the merger announcement on REIT acquirers' stock price returns. We followed the market model approach to assume a linear relationship between the expected return on a security and the return on the market portfolio. Specifically, for each security $i$, the market model assumes the return on security, given by:

$$R_{it} = \alpha_i + \beta_i R_{mt} + \varepsilon_{it}, \tag{1}$$

where $R_{it}$ is the return on security $i$ at time $t$. $R_{mt}$ is the return on the market portfolio at period $t$. The linearity and normality of returns are assumed, and $\varepsilon_{it}$ is the error. $\alpha_i$ and $\beta_i$ are coefficients. The market model expressed in Equation (1) is used to compute the expected return on the stock on the day of the event or during a selected event window. Equation (1) is first estimated with the sample observed during the 89-day estimation window from $t = -94$ to $t = -6$, where $t = 0$ is the event day.

The abnormal return (*AR*) due to the announcement on any given day, therefore, equals the actual return minus the predicted normal return:

$$AR_{it} = R_{it} - (\alpha_i + \beta_i R_{mt}). \tag{2}$$

To obtain a general insight into the abnormal return observations for a sample of *N* firms, the average abnormal returns (*AAR*) for each day *t* are averaged as follows:

$$AAR_t = \frac{1}{N} \sum_{i=1}^{N} AR_{it}. \tag{3}$$

The event window is the period between $\tau$ days prior to the event and $\tau$ days after the event. The expected returns on the stock calculated from Equation (1) for the security during the event window $(-\tau, +\tau)$ are compared with the actual returns on each day in the event window. The cumulative difference between the predicted return and the actual return in the event window is called the cumulative abnormal return and is calculated as follows:

$$CAR_i(-\tau, +\tau) = \sum_{t=-\tau}^{+\tau} AR_{it} \tag{4}$$

The last step is to calculate the *t*-value. First, the standard deviation *(S)* is calculated as follows:

$$S = \sqrt{\frac{\sum_{i=1}^{N}(CAR_i - CAAR_i)^2}{N - 1}}, \tag{5}$$

where $CAAR_i$ is the average value of $CAR_i$, and *N* is the total number of firms. Then, the *t*-value is calculated as follows:

$$t = \frac{CAAR_i - 0}{S/\sqrt{N}}, \tag{6}$$

where *N* is the total number of firms.

### 3.2.2. Cross-Sectional Regression Models

A cross-sectional regression was applied to identify the sources of cumulative abnormal returns from merger announcements. The dependent variable is the 2-day *CAR* (0, +1) for all regression models. The independent variables in the regression models are listed and illustrated in Table 3.

**Table 3.** Definition and summary statistics for the explanatory variables.

| Variable | Definition | Mean | Std dev. |
|---|---|---|---|
| *DOMES* | Type dummy, equals one for domestic M&A and zero otherwise. | 0.87 | 0.34 |
| *CRISIS* | Subprime dummy, equals one for M&As made during 2007Q2 and 2009Q1, and zero otherwise. | 0.33 | 0.47 |
| *DOMES*CRISIS* | Interacting dummy, equals one for domestic M&As made during subprime and zero otherwise. | 0.29 | 0.45 |
| *STATUS* | Target public status dummy, equals one for privately held and zero otherwise. | 0.66 | 0.47 |
| *STRUCTURE* | Method of payment dummy, equals one for all cash-financed and zero otherwise. | 0.44 | 0.50 |
| *R_SIZE* | Deal value divided by market capitalization of acquirers. | 0.09 | 0.22 |
| *ROE* | Return of equity from acquirers. | 0.08 | 0.13 |
| *EQUITY* | Equity divided by total assets from acquirers. | 0.38 | 0.18 |
| *STATES* | The number of states that the REITs firm has properties in. | 13.84 | 11.07 |

182 domestic and cross-border REITs M&A announcements from October 2005 to December 2010.

The major independent variable is domestic merger (*DOMES*), which is a dummy variable equaling one for a domestic merger and zero otherwise. If the cross-border merger announcements

consist of additional information, the coefficient of *DOMES* is expected to be statistically significant. The regression models control for the other acquisition attributes: public or private target (*STATUS*), payment structure (*STRUCTURE*), size of the transaction relative to the acquiring REIT (*R_SIZE*), return on equity of the acquiring REIT (*ROE*), the ratio of equity to total assets (*EQUITY*), and the number of states in which the acquiring REIT has properties (*STATES*).

STATUS equals one if the target REIT is privately held and zero if it is publicly held. It is expected to positively influence the abnormal return because privately held firms are frequently controlled by fewer investors who are easier to negotiate with than those in publicly held firms [30]. Also, in the corporate finance literature, Chang [8] concluded that the acquirer achieves positive abnormal returns in the announcement period of public-private mergers when the transaction is stock-financed. The wealth gain is argued to come from the monitoring by blockholders and the reduced information asymmetry.

STRUCTURE is also a dummy variable that controls for the method of payment of merger deals. STRUCTURE equals one if the merger deal is cash-financed and zero otherwise. It is expected to positively influence the abnormal return because acquirers prefer a cash payment when their stock is undervalued [31]. Previous studies, such as Campbell et al. [7], concluded that the acquiring REIT achieves positive abnormal returns during the announcement period in public-private mergers when the transaction is stock-financed. Although our data do not contain information on whether the merger transaction was stock-financed, we incorporated the cross-product term, $STATUS \times STRUCTURE$, to examine whether the acquirer will experience a positive or negative effect in a cash-financed merger transaction when the target REIT is private. R_SIZE is the relative value of both the target and acquiring firms. It is expected to positively influence the abnormal return due to the value-maximizing behavior exhibited by the management of bidding firms [4].

The variable *ROE* measures the acquirers' profitability and is expected to positively affect the abnormal returns, because the bidding firms with better profitability are better equipped to restructure the target firms [27]. *EQUITY* measures the capital strength of acquirers and is expected to have a positive influence because more leveraged firms are susceptible to a greater degree of investor sentiment [32].

STATES is a proxy variable of geographic diversification. Geringer et al. [33] considered the ratio of a company's foreign subsidiaries' sales to its total worldwide sales as the internationalization variable. Kim et al. [34] measured the degree of global diversification by the number of employees in foreign countries. Due to the characteristics of REIT firms having properties in states other than their asset portfolio, the number of states in which a REIT has properties was used as a proxy for the degree of geographic diversification. The cross-sectional regression model of the 2-day CAR is characterized as follows:

$$
\begin{aligned}
CAR\,(0,+1) = \beta_0 &+ \beta_1 DOMES + \beta_2\,STATUS + \beta_3 STRUCTURE \\
&+ \beta_4 STATUS \times STRUCTURE + \beta_5 R_{SIZE} + \beta_6 ROE \\
&+ \beta_7 EQUITY + \beta_8 STATES + \varepsilon
\end{aligned}
\tag{7}
$$

## 4. Empirical Results

### 4.1. Abnormal Returns of the Acquiring REITs in Domestic and Cross-Border Merger Announcements

Table 4 shows the *CAAR* around the merger announcements. *CAAR*s are reported for the three different event windows: *CAAR* (−1, 1), *CAAR* (0, 1), and *CAAR* (0, 2). Overall, the average return from all merger announcements was 0.8% surrounding a 3-day window (from day 0 to + 2) and 0.73% between day 0 and 1.

Our results are consistent with Allen and Sirmans [6], in which the acquirer had a positive abnormal return around the merger announcement. Campbell et al. [35] also found positive excess return for REIT bidders for the 3-day announcement period following public-private mergers. Overall, our findings support previous evidence in the real estate literature that indicate that bidding firms

enjoy more excess in their returns in mergers compared with findings from general corporate finance literature.

**Table 4.** Cumulative average abnormal returns for the acquiring trusts in REIT merger announcements.

| Day/window | Domestic M&A | Cross-Border M&A | All M&A Announcements |
|---|---|---|---|
| CAAR (−1,1) | 0.67%** | 0.21% | 0.61%** |
| CAAR (0,1) | 0.78%** | 0.41% | 0.73%** |
| CAAR (0,2) | 0.82%* | 0.66% | 0.80%** |
| Obs. | 160 | 22 | 182 |

** and * indicate significant at 5% and 10% significance levels, respectively. *CAAR* denotes the average value of the cumulative abnormal return (*CAR*).

Table 5 shows the mean cumulative abnormal returns for the acquiring trusts in domestic and cross-border REIT merger announcements from day 0 to day 1, *CAAR* (0, 1), during the three subperiods: before-the-crisis, during-the-crisis, and after-the-crisis periods. Only the *CAAR* for domestic merger announcements during the crisis period is positive and significant. Our results contrast with those reported by Amewu [36] and Beltratti and Paladino [27], who concluded that there was no significant change in abnormal returns on bidding firms' shares following the onset of the global financial crisis. The reason for the positive and significant abnormal returns for the acquiring REITs could be the value gains from investing in the distressed stock price of the target REITs during the crisis period.

The short-term wealth effects of the acquirers around the announcements of the domestic and cross-border mergers have also been documented in the corporate finance literature. Moeller and Schlingemann [15] provided evidence that U.S. firms who acquire cross-border targets relative to those that acquire domestic targets experience significantly lower announcement stock returns of approximately 1%. Acquirers' stock returns are negatively associated with global and industrial diversification. They concluded that the bidder return is positively associated with the legal system favoring strong shareholder rights and negatively associated with restrictive target countries. Mateev and Andonov [37] also found that bidding firms engaging in cross-border bids suffer lower announcement effects than those undertaking domestic acquisitions. They also provided evidence that cross-border bidding firms tend to suffer lower returns when the targets are located in countries with stronger investor protection mechanisms.

Furthermore, Table 5 shows that the average acquirers' *CAAR* was larger for cross-border merger announcements than that for domestic merger announcements before the crisis period, although neither is significant. During the crisis period, acquirers' *CAAR* was positive and statistically significant for domestic merger announcements only. In the aftermath of the crisis period, the acquirers' *CAAR* became smaller and insignificant for domestic mergers and negative for cross-border mergers, although neither is statistically significant.

We can summarize the findings for the short-term wealth effect of the REIT acquirers as follows. First, the acquirers achieved higher returns for domestic mergers than cross-border mergers. Acquirers realize a lower wealth effect from cross-border mergers. This is consistent with the argument in the literature that when the targets are in countries with more economic restrictions, such as investor protection mechanisms, which is likely true in general when compared with the U.S., the U.S. acquirers suffer lower returns than the returns in the case of domestic mergers. Second, acquirers achieved positive and statistically significant abnormal returns during the crisis period only. The gains in wealth were mainly from the distressed prices of the targets.

**Table 5.** Cumulative average abnormal returns for domestic and cross-border merger announcements during subperiods.

|  | Before the Crisis | | During the Crisis | | After the Crisis | |
|---|---|---|---|---|---|---|
|  | *CAAR*(0,1) | *t*-stats | *CAAR*(0,1) | *t*-stats | *CAAR*(0,1) | *t*-stats |
| Domestic | 0.13% | 0.43 | 1.86% | 2.89 *** | 0.34% | 0.55 |
| Cross-border | 0.58% | 1.47 | 1.28% | 0.62 | −0.83% | −0.85 |
| All | 0.19% | 0.71 | 1.79% | 2.92 *** | 0.21% | 0.37 |

*** indicates significant at 1% significance level.

### 4.2. Cross-Sectional Analysis of Abnormal Returns

Table 6 shows the results of the regressions of *CAR* from day 0 to day 1 for the acquiring firm during the three subperiods: before the crisis, during the crisis, and after the crisis, respectively. Before the crisis, when the target was domestic, the acquiring REIT achieved lower announcement returns than when the target was cross-border. This is consistent with the finding in Sahin [9] and with the internationalization theory, in which gains are from diversification when businesses seek synergies from intangible assets, such as information-based assets [10,11].

Second, the *CAR* of the acquiring REIT was larger when the target firm was private and when the merger transaction was cash-financed. Campbell et al. [7] concluded that acquirer returns were positive in stock-financed mergers when the target is private. Our results indicate that when the merger was cash-financed, the acquirer returns were positive, regardless of the public or private status of the target. This can be explained by the undervalued stocks of the acquiring REITs [31]. This finding—that when the target was private, the acquirer announcement returns were positive regardless of the method of payment—needs to be explained. In the corporate finance literature, Chang [8] concluded the acquirer achieves positive abnormal returns in the announcement period of public-private mergers when the transaction is stock-financed. He concluded that the wealth gain comes from the monitoring by blockholders and the reduction in information asymmetry. Campbell et al. [7] argued that the acquiring REIT valuation gain from the merger announcement when the target is private is better explained by the signaling effect.

However, the signaling effect cannot explain why the acquirer achieves a positive wealth effect from the merger announcement when the merger transaction is not stock-financed. For corporate acquisitions, Conn et al. [30] concluded that both domestic and cross-border private acquisitions result in positive announcement returns. They discussed several potential explanations for positive acquisition announcement returns when the target is private. For example, the process of making private bids is less exposed to the public gaze, and the acquirer can end the negotiation without loss of face. Poor acquisition outcomes due to hubris are less likely when the target is private. Also, the stocks of the private target are at a discount because of the illiquidity and other trading frictions. When the public acquirer purchases the assets of the private target, the potential value gain of the private target can be realized by the management of the public acquirer.

Finally, the size of the merger relative to the capitalization of the acquiring firm was marginally negatively associated with *CAR*, which contradicts our expectation. When we incorporate the interaction term of *STATUS* and *R_SIZE*, the coefficient of *R_SIZE* remains negative but becomes insignificant. The coefficient of the above interaction term is positive but, again, is insignificant (to save space, it is not reported in Table 6).

During the crisis period, only the size of the merger transaction relative to the capitalization of the acquiring firm was positively associated with the *CAR*. This finding is consistent with the finding in Table 5, in which the reason for the positive and significant abnormal returns for the acquiring firms could be the value gains from investing in the undervalued stocks of the target REITs during the

crisis period. This is supported by the argument of Reddy et al. [26]. As the deal size becomes larger, the gains from the merger for the acquiring firm become larger.

During the after-crisis period, the acquiring firms' *CAR* was larger when the target was domestic compared with the cross-border target. This reverses the finding from the pre-crisis period, but this is consistent with the finding of Conn et al. [30], who concluded that both the announcement and long-run returns of cross-border mergers are lower than those of domestic mergers. That is, before the crisis, synergies from international diversification dominates the effects of the aforementioned cross-border cultural differences, corporate governance, and valuation differences. However, following the crisis, the latter effects dominate the synergies from international diversification. This may suggest that since market participants became more risk-averse after the onset of the crisis, the market reacted to cross-border merger announcements less favorably than domestic mergers due to uncertainty and information asymmetries.

Furthermore, unlike the case for the pre-crisis period, when the target firm was private, the *CAR* became smaller, although it was not statistically significant. This corresponds to the increase in risk aversion of the market participants following the onset of the crisis: private target firms had more information asymmetry, and the stock prices of the acquiring REITs reacted to the merger announcement less favorably.

In the after-crisis period, when the acquisition was cash-financed, like the case for the pre-crisis period, the *CAR* was larger than that for other methods of payment. Finally, the number of states in which the acquirer had properties also positively affected the acquirer's *CAR*. This again corresponds to the argument of the increase in market participants' risk aversion. If the acquirer's properties had more geographic diversification, the acquirer's announcement return was higher. This effect of the number of states in which the acquirer has properties had no effect on the acquirer's merger announcement return before the crisis or during the crisis periods.

**Table 6.** Cross-sectional analysis: regressions of cumulative abnormal returns from day 0 to day 1.

| Independent Variable | Before the Crisis | During the Crisis | After the Crisis |
|---|---|---|---|
| *DOMES* | −0.0109 (−1.44) | 0.0158 (0.82) | 0.0426 ** (2.11) |
| *STATUS* | 0.0145 ** (2.22) | −0.01687 (−1.20) | −0.0283 ** (−2.00) |
| *STRUCTURE* | 0.0092 * (1.66) | −0.01221 (−0.80) | 0.0215 * (1.83) |
| *R_SIZE* | −0.0124 * (−1.79) | 0.2101 ** (2.44) | −0.0238 (−0.44) |
| *ROE* | 0.0246 (1.17) | −0.0436 (−1.14) | −0.0566 (−0.89) |
| *STATES* | $-1.2 \times 10^{-5}$ (−0.05) | −0.0005 (−0.96) | 0.0011 * (1.91) |
| Constant | −0.0124 (−1.19) | −0.0051 (−0.20) | −0.0524 ** (−2.27) |
| Obs. | 65 | 59 | 54 |
| Adj. *R*-square | 0.02 | 0.07 | 0.11 |

Numbers in parentheses are t-statistics. ** and * indicate significant at 5% and 10% significance levels, respectively.

## 5. Conclusions

A stylized fact concerning corporate acquisitions is that bidding firms experience much smaller returns than the target firms around the acquisition announcement date. This study examined the short-term wealth effect of the merger announcement on the acquiring REIT. Both domestic and

cross-border mergers were considered. We tried to reconcile our findings of the wealth effect from a merger announcement on the acquiring REITs with the existing theories and hypotheses on mergers and acquisitions in the corporate finance literature.

When the sample period was divided into pre-crisis, crisis, and after-crisis periods, we found that the acquiring trusts achieved positive and significant abnormal returns around the acquisition announcement date for domestic mergers during the crisis period only. This finding supports the argument that the acquiring REITs took advantage of attractive asset prices during the crisis period [26].

The cross-sectional analysis shows that, before the crisis period, privately held targets and cash-financed deals were positively associated with abnormal returns for the acquiring trusts. Chang [8] argued that when public-private mergers are stock-financed, the acquiring firms achieve positive returns due to the effects of the monitoring hypothesis and the signaling hypothesis. However, our finding cannot be explained by these hypotheses, because acquiring REITs achieved larger abnormal returns when the acquisition was cash-financed rather than stock-financed. During the crisis period, the size of the merger transaction relative to the capitalization of the acquiring trust became positively associated with the abnormal returns to the acquiring trusts. This finding reinforces the argument of the undervalued assets of the target. For example, Reddy et al. [26] found that emerging countries increased their foreign acquisitions in developed countries because of the more attractive asset prices. Following the crisis, the acquiring trusts achieved larger abnormal returns from domestic mergers around the announcement date than that associated with cross-border mergers. Furthermore, if the target trust was privately-held, the acquiring trust achieved smaller abnormal returns around the merger announcement date. Finally, when the acquiring trust had properties in more states, the trust achieved more abnormal returns following the crisis period.

Overall, this study provides evidence which sheds light on the changes in investors' risk preferences after the subprime mortgage crisis. Before the crisis, the acquirer achieved a larger wealth effect from a cross-border merger announcement than that from a domestic merger. This indicates that the benefits resulting from international diversification dominated the losses resulting from cultural differences, investor protection, corporate governance, and valuation differences. During the crisis period, the acquiring REIT achieved larger abnormal returns from the advantage of the undervalued assets of the target REIT. After the crisis, investors became more risk-averse such that the losses resulting from cultural differences, investor protection, corporate governance, and valuation differences dominated the benefits resulting from international diversification. The acquiring REIT also achieved larger abnormal returns around the acquisition announcement date when the acquiring trust had properties in more states.

**Author Contributions:** Conceptualization, A.T.W. and Y.-H.L.; Methodology, A.T.W. and Y.-H.L.; Software, A.T.W.; Validation, A.T.W., Y.-H.L. and Y.-C.C.; Formal Analysis, A.T.W.; Investigation, Y.-H.L.; Resources, A.T.W. and Y.-H.L.; Data Curation, Y.-C.C.; Writing—Original Draft Preparation, A.T.W. and Y.-C.C.; Writing—Review & Editing, A.T.W.; Visualization, A.T.W.; Supervision, Y.-H.L.; Project Administration, A.T.W.; Funding Acquisition, Y.-H.L.

**Funding:** This research was funded by [Ministry of Science and Technology of Taiwan] grant number [MOST 107-2410-H-006-029-]. And the APC was also funded by [MOST 107-2410-H-006-029-].

**Conflicts of Interest:** The authors declare no conflict of interest.

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
