# Peer review of "An Analysis of Gains to US Acquiring REIT Shareholders in Domestic and Cross-Border Mergers before and after the Subprime Mortgage Crisis"

_sustainability, doi:10.3390/su10124586_

Round 1

Reviewer 1 Report

In my opinion, there are three main drawbacks of this paper:

1) the problem of global financial crisis is placed as a central issue (Authors study the problem with reference to pre, during and post crisis period). However, the paper does not refer to the crisis - reasons and consequences, contagion effect, as well as its correlation with the mortgage loans market. The paper would benefit a lot if Authors add at least paragraph (with adequate literature references).

2) literature references - the empirical part seems well developed, but in my opinion the theoretical background is very limited... The study needs more reference to the motives and consequences of mergers (the current references are very scarce in comparison to the available literature body in this context)

3) relevance to the journal issue - sustainablity. In my opinion Authors should motivate the study or address the consequences/relevance of results to the journal main theme - sustainablity.

Author Response

Responds:

Reviewer 1:

  We have added the first paragraph concerning the discussions of global financial crisis. Relevant references have been included. The concept of sustainability has been addressed at the end of the first paragraph as well.

In the early period of the 2000s, the U.S. was experiencing mild economic growth and rising housing prices when the interest rates are low. Because of the rising home prices later on, bankers are encouraged to make more loans for higher compensations and the quality of the mortgage loans started to deteriorate, especially when the mortgage rates started to rise. The first disruption of credit market can be dated back to August 7, 2007, when the French bank BNP Paribas suspended redemption of shares held in some of its money market funds (Mishkin, 2011). Furthermore, on September 15, 2018, the investment bank Lehman Brothers filed for the largest bankruptcy in the U.S. history for its loss in the subprime mortgage market. Reinhart and Rogoff (2009) stated that the aftermath of crisis shares three characteristics: the asset market collapses are prolonged, the banking crisis is associated with profound declines in output and employment, and the level of government debt tends to explode. The outbreak of the global financial crisis in 2008 has reshaped the landscape of the financial markets as well. Investors have become more risk averse and that has changed the corporate strategies in business activities. It is important to examine how the corporations adjust their mergers and acquisitions strategies after the crisis and to explore how the market participants perceive the acquisition activities. To keep corporations sustainable, acquirers must identify the right targets from the long-term perspectives. For example, are the prices of the targets appropriate during the normal periods or the crisis periods? Will the acquisitions help the corporations grow? If the acquisitions are favorable, the market participants will revalue the acquirer’s stock price.

Reviewer 2 Report

This paper examines the abnormal returns of acquiring REITs near the announcement of acquisitions before and after the subprime mortgage crisis. Analysis includes both domestic and cross-border mergers, and cross-sectional regressions are also performed to study potential causes of observations. 

Overall the analysis clearly laid out its objective, methodology, data, observations and conclusions. The presentation of the paper is also done fairly well and the overall quality is as good as it is.

Author Response

Thank  you for reviewer's valuable comments.

We have also done the spell check!

Reviewer 3 Report

Dear all,

I read the paper „An analysis of gains to US acquiring REIT  shareholders in domestic and cross-border  mergers before and after the subprime mortgage crisis” and I find it very well structured and very interesting for readers.

Regarding the econometric analysis, I have a little doubt regarding the sample dimension, especially for cross-border mergers. It is too small to sustain the results. In this regard I suggest authors to present a descriptive statistics regarding all variables utilized in the analysis and also a correlation table.

It is too difficult for me to understand the content of table nr 4, in the paper general context (before, during and after crisis). So, I suggest to eliminate a such position.

I consider very useful a presentation of all cross-sectional analysis elements. In table nr 6 the Adj R square is too weak to sustain the results, so it is necessary to be presented much more information regarding regression analysis.

So, I suggest authors to check all the methodology, to extend the statistical presentation and to enlarge the cross-sectional analysis regression.

Author Response

Responds:

Reviewer 3:

  REITs papers have limited data. For example, the sample in Allen and Sirmans (1987) has only 52 successful events. Sahin (2005) has 33 targets and 32 acquirers. Our study has 182 announcements.

  Regarding the R-squares in the cross-sectional analysis, we admit that the R-squares are not satisfactory. However, our independent variables complement the gaps in previous studies. For example, Sahin (2005) considers Fama-French three-factor model. Allen and Sirmans (1987) did not conduct the cross-sectional analysis. Unlike the corporate M&A, data of REITs are less comprehensive.

Allen, P. R., and Sirmans, C. F., 1987. An analysis of gains to acquiring firm’s shareholders: The special case of REITs. Journal of Financial Economics 18, 175-184.

Sahin, O., 2005. The performance of acquisitions in the real estate investment trust industry. Journal of Real Estate Research 27, 321-342.
